# Ensemble of Deep Recurrent Neural Networks for Identifying Enhancers via Dinucleotide Physicochemical Properties

**DOI:** 10.3390/cells8070767

**Published:** 2019-07-23

**Authors:** Kok Keng Tan, Nguyen Quoc Khanh Le, Hui-Yuan Yeh, Matthew Chin Heng Chua

**Affiliations:** 1Institute of Systems Science, National University of Singapore, 25 Heng Mui Keng Terrace, Singapore 119615, Singapore; 2Medical Humanities Research Cluster, School of Humanities, Nanyang Technological University, 48 Nanyang Ave, Singapore 639798, Singapore

**Keywords:** enhancer DNA, gene expression, ensemble deep learning, dinucleotide physicochemical properties, transcription factor, biocomputing, high performance

## Abstract

Enhancers are short deoxyribonucleic acid fragments that assume an important part in the genetic process of gene expression. Due to their possibly distant location relative to the gene that is acted upon, the identification of enhancers is difficult. There are many published works focused on identifying enhancers based on their sequence information, however, the resulting performance still requires improvements. Using deep learning methods, this study proposes a model ensemble of classifiers for predicting enhancers based on deep recurrent neural networks. The input features of deep ensemble networks were generated from six types of dinucleotide physicochemical properties, which had outperformed the other features. In summary, our model which used this ensemble approach could identify enhancers with achieved sensitivity of 75.5%, specificity of 76%, accuracy of 75.5%, and MCC of 0.51. For classifying enhancers into strong or weak sequences, our model reached sensitivity of 83.15%, specificity of 45.61%, accuracy of 68.49%, and MCC of 0.312. Compared to the benchmark result, our results had higher performance in term of most measurement metrics. The results showed that deep model ensembles hold the potential for improving on the best results achieved to date using shallow machine learning methods.

## 1. Introduction

Enhancers are short deoxyribonucleic acid (DNA) fragments of 50–1500 base pairs (bp) that assume an important part in the genetic process of gene expression in which a DNA segment is replicated into ribonucleic acid (RNA) [1,2]. Due to their possibly distant location of up to 1 Mbp relative to the gene on which they perform a regulatory action, enhancers are difficult to identify using early biochemical experimental methods such as DNA foot-printing [3]. Identifying enhancers could be done via experimental and computational studies. For example, an interesting work from Rhie et al. [4] used epigenetic traits to trace enhancer networks and identify them in breast, prostate, and kidney tumors. Later, Blinka et al. [5] applied genome-wide chromatin immunoprecipitation technique on sequencing data to identify transcribed enhancers. Xiong et al. [6] retrieved the information of 10 tissues and identified the functions of enhancers across them. They then found that enhancers associated with transcription factors (TF), enhancer-associated RNAs (eRNA) and target genes appeared in tissue specificity and function across different tissues. The latest work from Arbel et al. [7] identified enhancers by exploiting regulatory heterogeneity and they achieved a high-accuracy of 85% regarding this problem.

The rise of computation bioinformatics [8] offers a new approach using machine learning methods to solve the problem of identifying enhancers. Since the method of support vector machines was proposed in 1995 [9], it has remained a popular machine learning model for performing classification tasks. Liu et al. [10] employed support vector classification (SVC) in their ‘iEnhancer-EL’ predictor using an ensemble approach in which a total of 171 SVC models were trained. 6 of these models were selected using a clustering method and formed into a model ensemble for identifying enhancers and non-enhancers. Based on the same approach, Liu et al. also further identified weak versus strong enhancers using an ensemble of 10 selected models. To achieve good and accurate results, manual feature engineering was performed to obtain features which aided the SVC models in learning the classification task. Each DNA sample was converted to a vector using K-mer subsequences, subsequence profiles, and pseudo k-tuple nucleotide composition. Jia et al. proposed an improvement to ‘iEnhancer-2L’ [11] an earlier solution from Liu et al. by introducing more features for the SVC model using 104 features from bi-profile Bayes, 1 from nucleotide and 9 from pseudo-nucleotide composition [12]. The benchmark dataset of this work had been applied in later applications such as iEnhancer-PsedeKNC [13], EnhancerPred2.0 [14].

Various other shallow machine learning models have also been applied to solve the enhancer identification problem. Hiram et al. emphasized the role of manual feature extraction in their use of a single hidden layer time-delay neural network for identifying enhancers using chromatin signatures [15]. Rajagopal et al. [16] leveraged histone modification as a feature in their random forest model as enhancer identification is also based on chromatin signatures. Erwin et al. used multiple kernel learning [17], a variant of support vector machines while Bu et al. used three-layer deep belief network on DNA sequence compositional, histone modification and DNA methylation features [18]. Le et al. [19] used a FastText model to transform DNA sequences into vectors and performed supervised classification via support vector machine.

While shallow machine learning methods can achieve good results given useful features, deep machine learning models offer the promise of even better results without the need for manual feature engineering. Powerful deep networks can learn deep distributed representations comprising high-level features in the data through the progressive composition of many non-linearities in each layer of the deep network [20]. While it is possible for a shallow network to learn similarly with manually derived features, it will require a high number of hidden nodes in each of its few hidden layers. It has been proven that a deeper network of more layers can function using a smaller number of hidden nodes compared to a shallow model, achieving a higher statistical economy. Deep-learning-based models have been used to predict enhancers with promising performance, such as convolutional neural network [21] and bidirectional gated recurrent units [22].

In addition, feature engineering using DNA subsequences such as K-mers - the equivalent of n-grams in text-mining - used in Liu et al. did not capture the full sequence information residing in a DNA sequence. The similar gap had been also seen in the feature engineering of FastText vectors [19]. The time-delay network in Hiram et al., being a 1D convolution in the temporal dimension [20], similarly sampled only subsequences. In recurrent neural networks, deep networks offer an alternative which performs full sequence processing.

Deep networks thus hold the potential to achieve better performance than large shallower networks in the task of enhancer identification, a claim which this study will seek to substantiate by developing an enhancer predictor based on the deep recurrent neural network and its model ensembles.

## 2. Materials and Methods

### 2.1. Convolutional Neural Networks

Convolutional neural networks [23] are best known for being useful in image classification tasks. In addition, they are also useful in processing sequential data [24]. In this context, the convolution layers are one-dimensional (1D) and each convolution operation employs a 1D window and filter which slides over each position in a sequence to extract sub-sequences. 1D convolutional networks learn local patterns in a sequence, which is similar to how convolutional 2D networks learn local patterns in an image. Down-sampling sequential data is performed via a 1D max pooling operation, which is equivalent to 2D max pooling for images. In this study, 1D convolution and max pooling layers were used to pre-process an input sequence into shorter sequences of higher-level features for the recurrent neural network [25]. Figure 1 shows such a model used in the study.

Using convolution combined with a recurrent network architecture allows the amount of sequence information used in the classification task to be controlled by tuning the convolution/max pooling layers.

### 2.2. Recurrent Neural Networks

Recurrent neural networks (RNN) are specialized neural networks for processing sequential data [20]. A RNN processes an input x by combining it with its hidden state h which is passed forward through time to the next input in a sequence in Equation (1).
(1)$$h^{\left(t\right)}=f\left(h^{\left(t-1\right)},\;x^{\left(t\right)},\;\theta \right)$$

In the above, the same value of θ parameterizes f for all the time steps. A typical RNN will have an output layer which takes information out of the hidden state to make predictions.

However, such a simple RNN performs poorly in the learning of long-term dependencies between the inputs and the final predicted output in long sequences. In 1997, [26] introduced the long short-term memory (LSTM) unit, a gated RNN which greatly improves the performance of RNNs in learning such dependencies. The gates in the LSTM unit are themselves fully-connected feed-forward neural networks. Proposed more recently by Cho et al. [27], the gated recurrent unit (GRU) is an RNN hidden unit selected in this study for its simplicity to implement and to compute in comparison with the LSTM. The GRU consolidates the three inputs, output and forget gates [28] in the complex LSTM architecture, whose design rationale is not totally clear as noted by Josefowicz et al. [29], into two reset and update gates while retaining an ability to learn dependencies over different time scales. Figure 2 shows a model with two GRU layers used in the study.

### 2.3. Model Ensembles

An ensemble of classifiers is a set of classifiers whose individual predictions are aggregated typically by a majority voting scheme to produce a final prediction on an input sample [30]. For an ensemble to produce better results than its individual members, each member must produce overall accurate results and different errors on input samples. Typical strategies for assembling machine learning models are bagging and boosting which require training multiple models by varying the training dataset or the sample weighting scheme respectively. For deep RNNs, these approaches are usually not feasible given the relatively long model training duration in comparison to shallower machine learning models such as SVCs.

In this study, to obtain the individual models of the ensemble, the technique of warm restarts is used in model training [31]. Warm restarts utilize a cyclic learning rate based on a simple cosine annealing algorithm. In each cycle, the learning rate starts at a maximum and decrease to a minimum over the length of the cycle according to Equation (2).
(2)$$\gamma_{t}=\gamma_{min}+\frac{1}{2}\left(\gamma_{max}-\gamma_{min}\right)\left(1+\cos\left(\frac{T_{cur}}{T}\pi \right)\right)$$

In Equation (2), γ is the learning rate, T is the number of epochs in a cycle, Tcur is the current epoch.

Warm restarts help in improving the rate of convergence during model training. In addition, it allows the harvesting of a model in each cycle of a training session [32]. During each cycle, the training converges to a local minimum at which the model weights correspond to a relatively good model. At the end of each cycle, resetting the learning rate back to maximum kicks the training trajectory out of the local minimum so that it will re-converge into a different local minimum in the next cycle. While the models at different minima may have similar error rates, they tend to make different errors which satisfy the condition for a model ensemble.

## 3. Experiment

### 3.1. Dataset and Machine Learning Task

The dataset used in this study was taken from Liu et al. [11] and subsequently also used in later published works [10,12,13,14]. It was perfectly balanced across classes and did not have any missing data. The training data of 2968 samples comprised 1484 samples each of enhancers and non-enhancers for the layer 1 binary classification problem. An independent test set of 200 samples of each class was used for evaluation. This dataset also contains a layer 2 binary classification problem of identifying strong or weak enhancers. The training data comprised 742 samples of each class. An independent test set of 100 samples of each class was used for evaluation.

According to Liu et al. [11], the dataset was a combination of the chromatin state information of nine cell lines. To obtain fragments consistent with the length of nucleosome and linker DNA, the data was separated into 200 bp fragments. Each data sample was thus a 200-element sequence of the 4 nucleotide bases ‘A’, ‘C’, ‘G’ and ‘T’. An example of a data sample was ‘CACAATGTAGA …’. Each dinucleotide or pair of nucleotide bases was also associated with a physicochemical property given in Table 1.

In [10], it has been noted that for evaluating the performance of predictors, three methods were commonly used. These were the independent dataset test, the subsampling test, and the jackknife test. In our study, we have opted to rely mainly on the first, the independent dataset test. Due to the long model training duration for RNNs (>5 h with regularization), employing subsampling or jackknife resampling required more resources than was available. As an example, performing subsampling using five-fold cross-validation would require ~25 h of training time per network architecture given that five separate models were trained in this process. In place of this, a simple train/validation split was used in our model training process. The difference in using a simple train/validation split lies in the less accurate estimates of model error in comparison to the more robust approaches. While this would affect the model selection process to obtain the best model, it does not compromise the final evaluation of the selected model, which is based on the independent dataset test [10,11].

### 3.2. Assessment of Predictive Ability

The benchmarks on the independent test set also were obtained from [11] which were believed to be the best results obtained to date from the dataset. As noted in that study, the most important metrics were accuracy and Matthews Correlation Coefficient (MCC) which determined how useful a model was. Besides that, they also reported the sensitivity and specificity. Finally, the metrics included accuracy (Acc), sensitivity (Sens) or recall, specificity (Spec), and Matthews Correlation Coefficient (MCC) defined as:(3)$$Sensitivity=\frac{TP}{TP+FN}$$
(4)$$Specificity=\frac{TN}{TN+FP}$$
(5)$$Accuracy=\frac{TP+TN}{TP+FP+TN+FN}$$
(6)$$MCC=\frac{TP*TN-FP*FN}{\sqrt{\left(TP+FP\right)\left(TP+FN\right)\left(TN+FP\right)\left(TN+FN\right)}}$$
where true positive (TP) and true negative (TN) are the numbers of correctly predicted enhancer and non-enhancer, respectively; whereas false positive (FP) and false negative (FN) are the numbers of misclassified enhancer and non-enhancer, respectively. These metrics have been used in a lot of recent bioinformatics problems to evaluate the performance [33,34]. In addition to that, we utilized the Receiver Operating Characteristics (ROC) curve and Area Under Curve (AUC) [35] as an indicator to evaluate the performance of our models. The AUC value provides a probability value ranging from 0 to 1. The greater the ROC curve deviates from the 45-degree diagonal, the more propinquity to the point (0, 1), and the greater the AUC value, the better the prediction performance.

### 3.3. Model Training Setup

The models were implemented using the Keras [36], Tensorflow [37] and Sci-kit Learn [38] machine learning libraries. Model training was performed mainly on Google Co-laboratory [39] with some base models trained on a stand-alone personal computer. Presently, RNN implementations on Keras/Tensorflow using dropouts [40] are not supported by NVIDIA Compute Unified Device Architecture (CUDA) Deep Neural Network (cuDNN) library [41]. This resulted in long model training duration of around 5 h or more when using RNNs with dropouts. The loss function and metric used were binary cross-entropy and accuracy respectively.

### 3.4. Feature Extraction

Each nucleotide base in a sample was first converted using one-hot encoding to a vector of shape (4,). Secondly, each of the di-nucleotide physicochemical properties in Table 1 was normalized to a range of 0 to 1. Each di-nucleotide was then converted to each of its 6 normalized physicochemical properties, giving a vector of size 6. As the last nucleotide base in each sample was not associated with any di-nucleotide, a value of −1 was used as padding. It has been shown that such physicochemical properties have a major role in gene expression regulation and they correlate with functional elements such as enhancers [42]. These physicochemical properties had been successfully used in numerous bioinformatics applications with high performance [10,43]. The state-of-the-art work on enhancer classification problem has also integrated physicochemical properties in their features [10]. Therefore, these features were used in our study.

In summary, each sample, comprising of a sequence of 200 nucleotide bases, was pre-processed into a vector of shape (200,10).

## 4. Results

### 4.1. Base Models

Different network architectures were implemented in the base models including both single and bi-directional GRU and convolution1D with max-pooling layers. The base models were scaled-up to increase their predictive power until overfitting was observed. In Table 2, the smallest model is #1 which has only a single GRU layer. The layer was increased in size in model #2. Additional GRU and dense layers were added as the models were gradually scaled-up. More complex layers which include bi-directional GRU and convolutional 1D layers were added in the later models. It can be seen that the best model performance was observed in the smallest model #1 at epoch 586 while in model #6, it was observed at epoch 44. This showed that more powerful models overfit earlier during model training. Overfitting caused the performance of larger models to deteriorate over their simpler counterparts. To realize the potential of these more powerful models, the detrimental effects of overfitting must be countered by applying regularization techniques in the next phase of model fine-tuning.

The activation function for all intermediate layers was relu except for GRU layers where default values in Keras of tanh and sigmoid were used. The optimizer used was RmsProp [44] with the default learning rate of 0.001. This same optimizer was also used in model fine-tuning and for the model ensembles with varying learning rates.

None of the base models (in Table 2) were observed to outperform the benchmark. Two models 4 (Figure 2) and 7 (Figure 1) with different network architecture were selected for further fine-tuning and regularization.

### 4.2. Model Fine-Tuning

Dropouts with different parameter values and weights regularization [45] were applied to two selected base models (#4, #7). As larger networks are typically viable with dropouts, the GRU layers of the base model #7 were also enlarged from 8 to 16 hidden nodes (Table 3). It was observed that accuracy increased and overfitting set in only later in the training process.

### 4.3. Model Ensembles

The hyper-parameters used for training both selected models with warm restarts are shown in Table 4. From each training run, the best model was saved every 200 epochs or cycle. Several model ensembles, each comprising different combinations of individual models, were scored on the validation split.

For layer 1, based on validation accuracy, 5 models were finally selected from the cycles between epochs 800 and 1800 which includes the best model from epoch 1283 for model ensemble #1 (Table 3). For model ensemble #2, 3 models from the cycles 200–400, 800–1000 and 1400–1600 were selected. An example of the cycles in model training with warm restarts is seen in Figure 3. Warm restarts utilize a cyclic learning rate based on a simple cosine annealing algorithm. In each cycle, the learning rate starts at a maximum and decrease to a minimum over the length of the cycle according to Equation (2). During each cycle, the training converges to a local minimum (corresponding to the local peaks in training accuracy) at which the model weights correspond to a relatively good model. At the end of each cycle, resetting the learning rate back to maximum kicks the training trajectory out of the local minimum (corresponding to the local sharp dips in training accuracy) so that it will re-converge into a different local minimum in the next cycle (refer to 2.3 Model Ensembles). The warm restart cycles are apparent in the training accuracy plot where each repeated “sawtooth” pattern is a cycle. A similar model selection process was performed for the layer 2 model ensembles.

Base on the validation results in Table 5, the best models were model ensemble #1 and #3 for two layers, respectively.

### 4.4. Cross-Validation Results

Since previous methods used cross-validation strategy in their model, we aimed to evaluate our cross-validation performance in this section. Table 6 shows our 5-fold cross-validation results in both layer 1 and layer 2 of enhancer classification. As shown in this table, our performance results were consistent and did not contain many differences compared to Table 5. It means that our model could be applied via different evaluation methods (train/test splitting or cross-validation) with a stable performance.

### 4.5. Independent Test Results

Independent test is an utmost step to evaluate whether machine learning models perform well in both seen and unseen data. It was used to check the correctness as well as overfitting of every model. Since Liu et al. provided benchmark datasets including the independent dataset (200 enhancers and 200 non-enhancers), we also used these independent datasets in this section. The best single models after regularization and the model ensembles were evaluated on this set. Table 7 shows that our independent results are at the same level with validation results. Meanwhile, there is no overfitting in our models.

ROC Curves were provided in this section to see our performance results at different threshold levels. As shown in Figure 4, the ensemble model helps us reach the best performance in the first layer, but it cannot perform well in the second layer. For layer 2, the best model was model #6 (Single Model 1 × 16 Conv1D, 2 × 16GRU-B, and 1 × 8 Dense).

## 5. Discussion

### 5.1. General

Table 8 shows the comparative performance between our results and the other results on the same independent dataset. As noted in Liu et al. [11], while a broad range of metrics were calculated for comparison, the most important metrics are accuracy which reflects the predictor’s overall performance and MCC, its stability. Therefore, we also showed the detailed comparison of these metrics in Figure 5.

For layer 1, both model ensembles outperformed the benchmark from iEnhancer-EL [10] as well as other predictors for test accuracy and MCC. The model ensembles also outperformed the single models. The best model comprised a model ensemble using as a base learner the model with only GRU layers with 5 individual models in the ensemble (#4 in Table 7) which achieved a test accuracy of 75.5% and MCC of 0.510 against the benchmark values of 74.75% and 0.496 respectively. For layer 2, the best model comprised the single model with both convolution 1D and bi-directional GRU layers (#6 in Table 7) which achieved a test accuracy of 68.49% and MCC of 0.312 against the benchmark values of 61.00% and 0.222 respectively. It is easy to see that we could improve performance a lot in layer 2 when compared with the other previous works. The model ensembles did not perform better than the single models, however, there is a need for further testing different cases of ensemble. Here, an important discussion is that our model did not outperform the previous works on enhancers such as DeepEnhancer [21] and BiRen [22]. Although both of them also identified enhancers using DNA sequence and deep neural network, but they used a different dataset compared with us. Therefore, to perform a fair comparison, further studies could apply our network on different datasets. Also, the other works could not classify enhancers into strong and weak enhancers. It is also an essential problem and it explains the reason why we selected the dataset of Liu et al. [11] in this study.

In the present study, to simplify the model training process, the same network architecture as the layer 1 classifier was employed for the layer 2 classifier. The same selection of individual models was also made for the model ensembles. As such, the network architecture as well as the model selection in the model ensembles were not optimized for the layer 2 problem. This is likely the reason for the underperformance of the model ensembles relative to the single models. Better performance could possibly be obtained by going through a model selection process via base models and fine-tuning to select an optimal network architecture for the layer 2 classification problem. A separate model selection for the model ensembles could also be performed. This additional work could be attempted in a later study.

Moreover, to demonstrate generality, we used our model to predict another dataset on enhancer. The dataset used in this section was collected directly from [46], in which the authors retrieved 900 enhancer sequences and 898 non-enhancer sequences from VISTA Enhancer Browser. We used this dataset as an independent dataset and inserted into our model. For the results, our single model outperformed the ensemble model with achieved accuracy of 56.007% and MCC of 0.125. Compared to our benchmark independent dataset, there was a little bit lower because of some explainable reasons. First of all, the test performance is highly dependent on the test dataset. If a dataset has more of the same kind of sample as in the benchmark dataset, the overall performance will be poorer than what we obtained on Liu et al. [10]. Secondly, there are differences in the length of DNA samples between two datasets (Liu et al. [10] used 200 bp and used 1500 bp, thus we cannot use a model with length of 200 bp to predict the sequences with length of 1500 bp. Therefore, if we can train a model with the pure dataset from [46], the performance should be better. We hope to find an optimal model that could be applied in both datasets with a promising performance in the future works.

### 5.2. Model Ensembles

Model ensembles have been shown to yield better results given sufficient diversity and quality in its component models [30]. One common method of building model ensembles is bootstrap aggregation, also known as bagging. In bagging, diversity in the ensemble’s component models is achieved by a random re-sampling of the training data for each model. In this study, the need for training each component model on a different dataset was obviated by using the warm restart technique. It is noteworthy that the model ensembles are similar to an ensemble of bagged predictors according to [47]. Breiman deemed that 25–50 models in an ensemble of bagged models was a reasonable number. In this study, the number of models harvested was limited to less than 15 per training session using warm restarts (out of which 5 or less were finally selected) due to limitations in computational resources. Larger model ensembles can be explored in future work.

Breiman [48] also investigated another method for creating model ensembles commonly known as boosting [49]. In boosting, models were added progressively to the model ensemble where each model added depended on the errors made by the previous model. For example, in the popular Adaboost [50] method, samples wrongly classified by the previous model were weighted more before resampling the dataset for the next model. Breiman showed that boosting could yield improved results for larger model ensembles (of up to 250 models), as compared to bagging.

Specifically for model ensembles of neural networks, Moretti et al. applied bagging of fully-connected neural networks in their urban traffic flow forecasting model [51] while Khwaja et al. used the same approach in short-term energy load forecasting [52]. Several other studies employed boosted fully-connected neural networks using AdaBoost which produced occasional instances of exceptional results [53,54,55]. Mao [56] found that bagging out-performed boosting when metrics other than accuracy were employed. [57] applied boosting to RNN on a time-series regression task to produce more accurate results as compared to a single RNN model. With promising results obtained from these studies, future work should explore a boosting approach. As boosting is not amenable to parallelization, it necessitates an even longer model training duration as compared to bagging. As such, the essential computational resources will need to be secured for this endeavor.

### 5.3. Biological Discussion

In addition to the importance of deep neural network ensembles, biological features play an important role in classifying enhancers. The model performance had improved significantly when we added physicochemical properties to our neural network. It is because the ensemble neural network plays as a black box nature and it needs biological features to be learned. Since physicochemical properties had been used successfully in a lot of bioinformatics applications [42,43], this study provided more evidence on the usefulness of this feature in DNA sequence classification. However, further studies need to be developed to construct better hyperparameters in physicochemical properties to increase performance. Each enhancer or strong/weak enhancer sequence may have a different set of physicochemical properties and the optimal choice will help us increase the results. Moreover, understanding more features or nucleotide-binding sites that affect the classification of enhancers is an utmost important step for further works. Via this way, the machine learning algorithm is able to capture more information and possibly boost performance.

### 5.4. Source Codes and Web Application for Reproducing the Results

The python notebooks for pre-processing the data and training the various models are available in the following GitHub repository: https://github.com/tkokkeng/KE5006-AppliedResearch. We also deployed a web application namely DeployEnhancerModel which is freely available at https://github.com/tkokkeng/DeployEnhancerModel/. The users are able to access this link and run the web application without much computational knowledge. The instructions for running the web application is in the README.md file and as follows:
(1)Step 1: git clone this repository to your local machine using the command below:“git clone https://github.com/tkokkeng/DeployEnhancerModel”(2)Install all the required python package dependencies using the command below:“pip install -r requirements.txt”(3)Run the web server and Flask application using the command below:“python3 manage.py runserver”(4)Open a web browser and enter the following url: localhost:5000. The users should see our web application.

## 6. Conclusions

Identifying DNA enhancers is an important issue in biological aspects, especially computational biology. To address this issue, numerous computational researchers had conducted their classifiers with a variety of machine learning/deep learning techniques on different sets of features. In this study, a model ensemble of deep recurrent neural network base learners was shown to outperform the best results available-to-date for the first-layer enhancer identification task on the same dataset. Additionally, suggestions to improve the existing model ensembles for both the first and second-layer predictions were also proposed.

## Figures and Tables

**Figure 1 cells-08-00767-f001:**
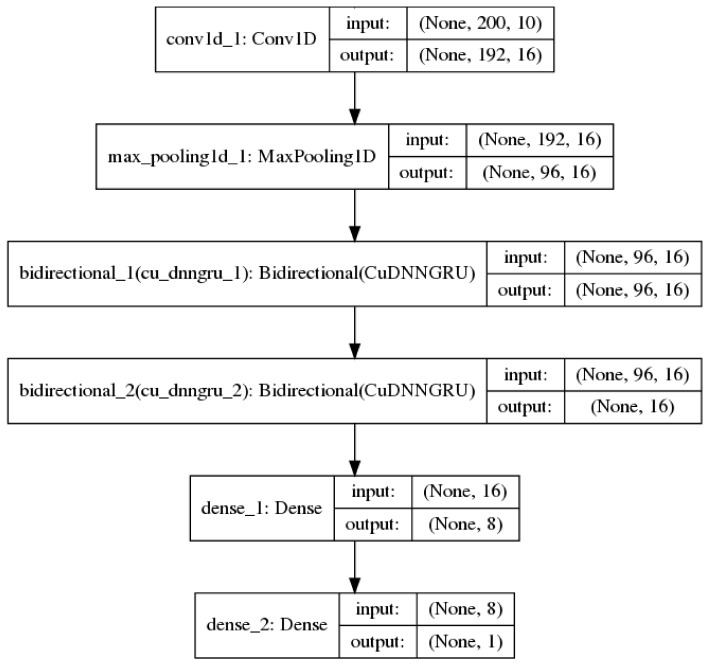
A neural network model with 1 convolution 1D and 1 max pooling layer before bi-directional recurrent and fully-connected layers.

**Figure 2 cells-08-00767-f002:**
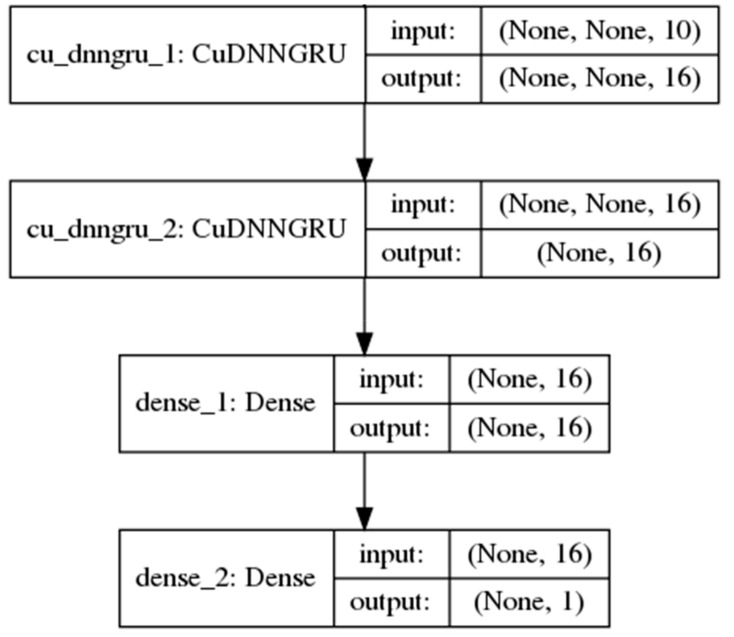
A neural network model with 2 single-direction gated recurrent unit (GRU) layers and 1 fully-connected layer.

**Figure 3 cells-08-00767-f003:**
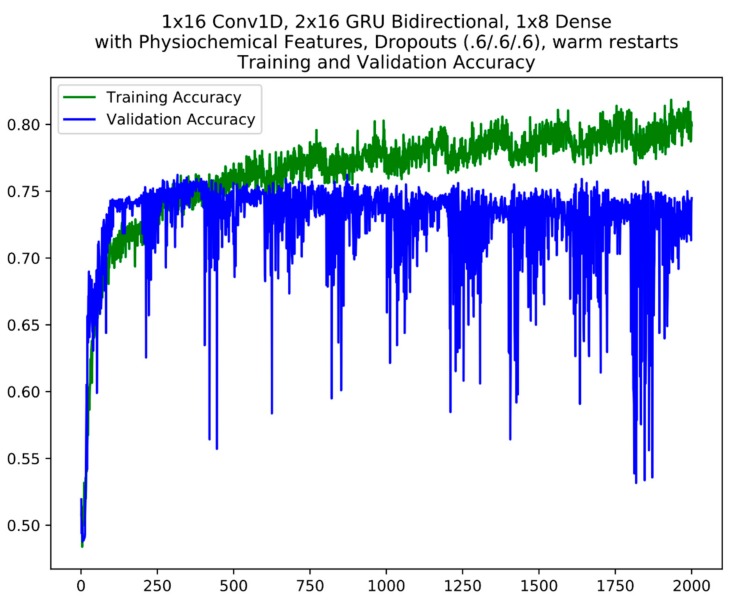
An example of model training using warm restarts. The cycles are apparent in the plot. The y axis is accuracy; the x axis is epochs.

**Figure 4 cells-08-00767-f004:**
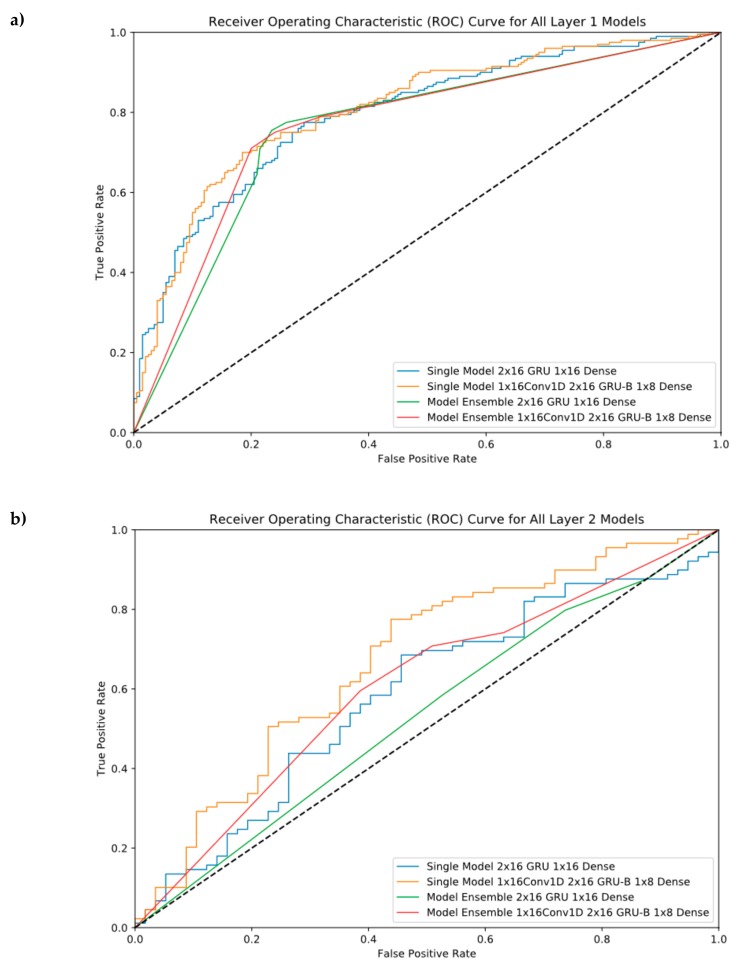
Receiver Operating Characteristic (ROC) Curve for all single and ensemble models. (**a**) Layer 1 classification, (**b**) Layer 2 classification.

**Figure 5 cells-08-00767-f005:**
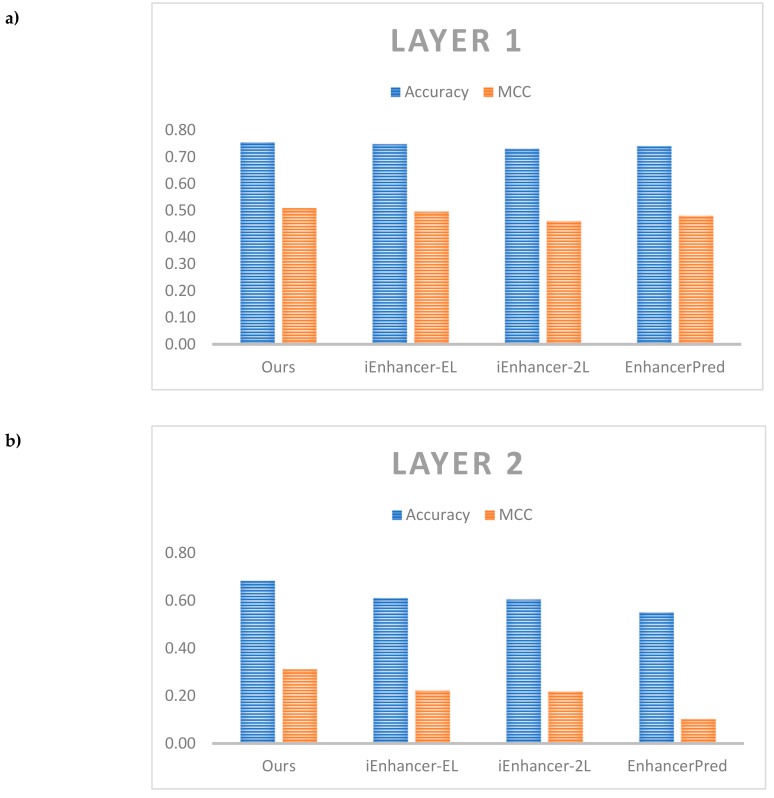
Comparative performance among different predictors. (**a**) Comparison on layer 1, (**b**) Comparison on layer 2.

**Table 1 cells-08-00767-t001:** Physicochemical property.

	Rise	Roll	Shift	Slide	Tilt	Twist
AA	0.430303	0.403042	1.000000	0.545455	0.4	0.833333
AC	0.818182	0.695817	0.618557	1.000000	0.7	0.833333
AG	0.257576	0.315589	0.762887	0.772727	0.3	0.791667
AT	0.860606	1.000000	0.319588	0.863636	0.6	0.750000
CA	0.045455	0.220532	0.360825	0.090909	0.1	0.291667
CC	0.548485	0.171103	0.731959	0.545455	0.3	1.000000
CG	0.000000	0.304183	0.371134	0.000000	0.0	0.333333
CT	0.257576	0.315589	0.762887	0.772727	0.3	0.791667
GA	0.706061	0.277567	0.618557	0.500000	0.4	0.833333
GC	1.000000	0.536122	0.494845	0.500000	1.0	0.750000
GG	0.548485	0.171103	0.731959	0.545455	0.3	1.000000
GT	0.818182	0.695817	0.618557	1.000000	0.7	0.833333
TA	0.000000	0.000000	0.000000	0.136364	0.0	0.000000
TC	0.706061	0.277567	0.618557	0.500000	0.4	0.833333
TG	0.045455	0.220532	0.360825	0.090909	0.1	0.291667
TT	0.430303	0.403042	1.000000	0.545455	0.4	0.833333

**Table 2 cells-08-00767-t002:** Model validation results.

S/N	Conv1D/Maxpool	GRU	Dense	Acc (%)	@Epoch
Layer 1					
#1	-	1 × 8	-	75.61	586
#2	-	1 × 16	-	75.20	252
#3	-	2 × 16	-	75.51	232
#4	-	2 × 16	1 × 16	75.41	392
#5	-	1 × 16b	1 × 16	74.82	206
#6	1 × 16(9)/2	2 × 8	1 × 8	74.49	44
#7	1 × 16(9)/2	2 × 8b	1 × 8	75.00	161
Layer 2					
#8	-	2 × 16	1 × 16	62.29	61
#9	1 × 16(9)/2	2 × 8b	1 × 8	60.27	68

Conv1D/Max pool = 1 × 16(9)/2 means 1 convolution layer of 16 channels using filter of size 9 and 1 max pool layer of filter size 2; stride is 1 for both GRU, Dense = 1 × 8 means 1 layer of 8 hidden nodes; b means bi-directional.

**Table 3 cells-08-00767-t003:** Fine-tuning validation results.

S/N	Conv1D/Maxpool	GRU	Dense	Regularization	Acc (%)	@Epoch
Layer 1						
#1	-	2 × 16	1 × 16	dp 0.1	77.65	865
#2	-	2 × 16	1 × 16	dp 0.2	76.22	1044
#3	-	2 × 16	1 × 16	dp 0.3	76.12	3744
#4	-	2 × 16	1 × 16	dp 0.4	75.10	1104
#5	1 × 16(9)/2	2 × 16b	1 × 8	dp 0.6	77.24	693
#6	1 × 16(9)/2	2 × 16b	1 × 8	dp 0.6; l1l2 3 × 10^−5^	77.14	604
Layer 2						
#7	-	2 × 16	1 × 16	dp 0.1	62.96	547
#8	1 × 16(9)/2	2 × 16b	1 × 8	dp 0.6	61.28	2081

Regularization = dp means dropouts with following parameter value; l1l2 means weights regularization using both l1 and l2 penalty functions with following parameter value. b means bi-directional.

**Table 4 cells-08-00767-t004:** Training (with warm restarts) validation results.

S/N	Conv1D/Maxpool	GRU	Dense	Warm Restarts	Acc (%)	@Epoch
Layer 1						
#1	-	2 × 16	1 × 16	dp 0.1; cyc 200; max_lr 0.003	76.73	1283
#2	1 × 16(9)/2	2 × 16b	1 × 8	dp 0.6; cyc 200; max_lr 0.001	76.22	871
Layer 2						
#3	-	2 × 16	1 × 16	dp 0.1; cyc 200; max_lr 0.003	63.30	228
#4	1 × 16(9)/2	2 × 16b	1 × 8	dp 0.6; cyc 200; max_lr 0.001	59.60	1224

Warm Restarts = dp means dropouts with following parameter value; cyc means number of epochs in a cycle; max_lr means maximum learning rate; minimum learning rate is set at 1 × 10^−4^ for both models. b means bi-directional.

**Table 5 cells-08-00767-t005:** Ensembles validation results.

S/N	Conv1D/Maxpool	GRU	Dense	# of Models	Acc (%)
Layer 1					
#1	-	2 × 16	1 × 16	5	77.45
#2	1 × 16(9)/2	2 × 16b	1 × 8	3	76.43
Layer 2					
#3	-	2 × 16	1 × 16	3	63.64
#4	1 × 16(9)/2	2 × 16b	1 × 8	3	59.60

# of Models = number of individual models in the model ensemble. b means bi-directional.

**Table 6 cells-08-00767-t006:** Cross-validation results.

Acc (%)	MCC	Sn (%)	Sp (%)	AUC (%)
Layer 1				
74.83	0.498	73.25	76.42	76.94
Layer 2				
58.96	0.197	79.65	38.28	60.68

MCC = Matthews Correlation Coefficient.

**Table 7 cells-08-00767-t007:** Independent test results.

S/N	Conv1D/Maxpool	GRU	Dense	Type	Acc (%)	MCC	Sn (%)	Sp (%)	AUC (%)
Layer 1									
#1	-	2 × 16	1 × 16	Single	74	0.48	75.00	73.00	79.63
#2	1 × 16(9)/2	2 × 16b	1 × 8	Single	73.75	0.475	75.00	72.50	80.86
#3	-	2 × 16	1 × 16	Ensemble	75.25	0.506	73.00	77.50	76.16
#4	1 × 16(9)/2	2 × 16b	1 × 8	Ensemble	75.5	0.51	75.50	76.00	77.04
Layer 2									
#5	-	2 × 16	1 × 16	Single	60.96	0.100	86.52	21.05	58.57
#6	1 × 16(9)/2	2 × 16b	1 × 8	Single	68.49	0.312	83.15	45.61	67.14
#7	-	2 × 16	1 × 16	Ensemble	58.90	0.071	79.78	26.32	53.19
#8	1 × 16(9)/2	2 × 16b	1 × 8	Ensemble	62.33	0.201	70.79	49.12	60.48

MCC = Matthews Correlation Coefficient. b means bi-directional.

**Table 8 cells-08-00767-t008:** Independent test results between our proposed method and the other state-of-the-art predictors.

Predictors	Acc (%)	MCC	Sn (%)	Sp (%)	AUC (%)
**Layer 1**					
Ours	75.50	0.510	75.5	76.0	77.04
iEnhancer-EL	74.75	0.496	71.0	78.5	81.73
iEnhancer-2L	73.00	0.460	71.0	75.0	80.62
EnhancerPred	74.00	0.480	73.5	74.5	80.13
**Layer 2**					
Ours	68.49	0.312	83.15	45.61	67.14
iEnhancer-EL	61.00	0.222	54.00	68.00	68.01
iEnhancer-2L	60.50	0.218	47.00	74.00	66.78
EnhancerPred	55.00	0.102	45.00	65.00	57.90

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
