# Peer review of "Ensemble of Deep Recurrent Neural Networks for Identifying Enhancers via Dinucleotide Physicochemical Properties"

_cells, 2019, doi:10.3390/cells8070767_

Round 1

Reviewer 1 Report

Tan et al. proposes a model ensemble of 16 classifiers for predicting enhancers based on deep recurrent neural networks. The model which uses this ensemble 19 approach could identify enhancers with achieved sensitivity of 75.5%, specificity of 76%, accuracy 20 of 75.5%, and MCC of 0.51.  In general, this paper looks more like a project report than a journal paper due to the reasons mentioned below; thus, I think that it should undergo substantial revisions before it is considered again for publication in a journal.

Major

-------

1) The paper is very short and most of it includes review of definitions that appear in text book or previous papers (e.g. RNN, sensitively, specificity, etc  ); thus, if I consider only the new information provided in the paper it is only ~5 pages long. A typical journal paper (in the field of this paper) should be at least 4 times longer.

2) Why did you work with 200 nt long sequences ? does this value optimized ? how the results change with this length ?

3) It was not clear why/how you choose  the topology of the network.  why not a different topology ?

4) The paper includes only 3 extremely low quality figures.  I expect to see at least 4 times more figures/sub-figures with details relate to the performances, comparison to other studies, biological discussion,   analysis of sub-datasets, etc...

5) Could you please explain the periodicity in the signals that appear in figure 3.

6) Comparison to previous approaches. You should compare the performances to at least 3 previous studies that were published on the topic in a Q1 journal in the last 3 years. If you do not convince the reader that the approach is indeed better than previous ones I do not see a reason to publish this study.  

Currently there is only comparison to [10] which is not very convincing (see comments 7)).

7) Table 7: it is not clear if the performances of the new algorithm are indeed significantly better than iEnhancer-EL. Please perform for example at least 20 Jackknife resampling iteration where you perform the analysis on subset of the date and compute a p-value related to the performances of the new method in comparison to the previous ones.

8) " We are making efforts to create a web server for front-end users. " Please indeed generate a website such that it will be possible to use the method (and also compare it to other methods). Without such a website the paper is not very useful.

9) I will be happy to see some discussion related to the biological data. For example, where the algorithm succeed and when it fails is it related to  the type/features of the enhancer , etc.   

10) what is the running time of the model?

11) it will be good to analyze at least one additional dataset to the one of Liu et al.

Reviewer 2 Report

The authors present a deep recurrent neural network model for enhancer prediction based on 6 dinucleotide physiochemical properties. They use the same dataset from Liu et al., 2018 and compare the performance of their model with several published models for predicting enhancers including the results in Liu et al, 2018. The recurrent neural networks proposed by the authors improved the prediction capacity evaluated by performance metrics such as accuracy, sensitivity, sensibility where this work adds to the field of enhancer prediction. 

Mayor concerns:

-In page 5 line 155: please explain the advantage or disadvantage of using a simple train/validation split instead of 5-fold cross validation used by other authors. 

-In page 5 lines 173-175 the authors wrote that they utilized the AUC of the ROC to evaluate the performance of the models. Yet, there are no results from this measure in the manuscript. Please provide the Area under ROC curve (AUC) for their model in Table 7 in order to compare it with the AUC from other state-of-the-art models provided in this table where the authors conclude their model outperforms the published ones. 

-Please discuss your results with the work of Min et al. 2017 Predicting enhancers with deep convolutional neural networks. BMC Informatics 18: 478 and Yang et al., 2017 BiRen: predicting enhancers with a deep-learning-based model using the DNA sequence alone. Bioinformatics 33: 1930.

-Please discuss the limitation that the model has a superior performance only for the 1-layer enhancer identification. 

Minor concerns:

In References section, Liu et al reference #10 is duplicated as #11. Please eliminate reference #11 in page 10 line 332.

Reviewer 3 Report

Tan et al. describe model ensemble of deep learning neural networks for identifying enhancers. They describe an interesting approach where physiochemical properties of dinucleotides can contribute to improved performance. 

The dataset is taken from a previous work and not much explanation is given otherwise.  The manuscript spends most of the time describing the neural network architecture and training methods and spend little time on explaining the importance of utilizing dinucleotide physiochemical properties (or could it be more than dinucleotide?). Also, it is not clear why the authors chose to use simple train/validation split rather than multiple cross-validation strategies. 

There is small improvement in terms of sensitivity and specificity (table 7), but it is not clear whether the small improvement over the existing predictors is enough to merit publication.  The authors need to test their model against other datasets to demonstrate generality.  

Thus, the manuscript seems more appropriate to a specialized journal such as Bioinformatics. 

Author Response

Reviewer #3

We greatly appreciate the reviewer #3 to carefully review the manuscript and offer valuable suggestions on our work. All of the comments and suggestions were useful for us to improve the quality of this manuscript. We have carefully addressed all fruitful comments in the below section. We feel that the revised manuscript is of considerably improved quality and hope that it can meet the increasingly high-quality standard of this journal. 

Comment 1:

The dataset is taken from a previous work and not much explanation is given otherwise.  The manuscript spends most of the time describing the neural network architecture and training methods and spend little time on explaining the importance of utilizing dinucleotide physiochemical properties (or could it be more than dinucleotide?). Also, it is not clear why the authors chose to use simple train/validation split rather than multiple cross-validation strategies.

Reply:

The reviewer is appreciated for the valuable comments. In this revised manuscript, we have explained more on dataset as well as dinucleotide physicochemical properties. Please check again in the whole manuscript and as follows:

“The dataset used in this study was taken from Liu et al [11] and subsequently also used in latter published works [10, 12-14]. It was perfectly balanced across classes and did not have any missing data. The training data of 2968 samples comprised of 1484 samples each of enhancers and non-enhancers for the layer 1 binary classification problem. An independent test set of 200 samples of each class was used for evaluation. This dataset also contains a layer 2 binary classification problem of identifying strong or weak enhancers. The training data comprised of 742 samples of each class. An independent test set of 100 samples of each class was used for evaluation.

According to Liu et al. [11], the dataset was a combination of the chromatin state information of nine cell lines. To obtain fragments consistent with the length of nucleosome and linker DNA, the data was separated into 200 bp fragments. Each data sample was thus a 200-element sequence of the 4 nucleotide bases ‘A’, ‘C’, ‘G’ and ‘T’. An example of a data sample was ‘CACAATGTAGA …’. Each dinucleotide or pair of nucleotide bases was also associated with a physicochemical property given in Table 1.”

“Each nucleotide base in a sample was first converted using one-hot encoding to a vector of shape [4,]. Secondly, each of the di-nucleotide physicochemical properties in Table 1 was normalized to a range of 0 to 1. Each di-nucleotide was then converted to each of its 6 normalized physicochemical properties, giving a vector of size 6. As the last nucleotide base in each sample was not associated with any di-nucleotide, a value of -1 was used as padding. It has been shown that such physicochemical properties have a major role in gene expression regulation and they correlate with functional elements such as enhancers [42]. These physicochemical properties had been successfully used in numerous bioinformatics applications with high performance [10, 43]. The state-of-the-art work on enhancer classification problem has also integrated physicochemical properties in their features [10]. Therefore, it is potential to use these features in our study.”

Moreover, to explain the reason why we chose to use simple train/validation split rather than multiple cross-validation strategies, we have inserted a paragraph in section 3.1, page 5 as follows:

“In [10], it has been noted that for evaluating the performance of predictors, three methods were commonly used. These were the independent dataset test, the subsampling test and the jackknife test. In our study, we have opted to rely mainly on the first, the independent dataset test. Due to the long model training duration for RNNs (>5 hours with regularization), employing subsampling or jackknife resampling required more resources than was available. As an example, performing subsampling using five-fold cross-validation would require ~25 hours of training time per network architecture given that five separate models were trained in this process. In place of this, a simple train / validation split was used in our model training process. The difference in using a simple train / validation split lies in the less accurate estimates of model error in comparison to the more robust approaches. While this would affect the model selection process to obtain the best model, it does not compromise the final evaluation of the selected model, which is based on the independent dataset test [10, 11].”

Comment 2:

There is small improvement in terms of sensitivity and specificity (table 7), but it is not clear whether the small improvement over the existing predictors is enough to merit publication.  The authors need to test their model against other datasets to demonstrate generality.

Reply:

The reviewer is appreciated for the valuable comment. In the revised version, we have also performed our experiments on layer 2 classification. It can be considered as a second dataset and our model also performed well on this dataset. Another important point is that we can also get a good result on both validation and independent dataset. Please check it again in our revised manuscript.

Table 7. Comparison of independent test results between our proposed method and the other state-of-the-art predictors

Predictors

Acc (%)

MCC

Sn (%)

Sp (%)

AUC (%)

Layer 1

Ours

75.50

0.510

75.5

76.0

77.04

iEnhancer-EL

74.75

0.496

71.0

78.5

81.73

iEnhancer-2L

73.00

0.460

71.0

75.0

80.62

EnhancerPred

74.00

0.480

73.5

74.5

80.13

Layer 2

Ours

68.49

0.312

83.15

45.61

67.14

iEnhancer-EL

61.00

0.222

54.00

68.00

68.01

iEnhancer-2L

60.50

0.218

47.00

74.00

66.78

EnhancerPred

55.00

0.102

45.00

65.00

57.90

Round 2

Reviewer 2 Report

No more comments.

Author Response

We greatly appreciate the reviewer to carefully review the manuscript and offer valuable suggestions on our work. All of the comments and suggestions were useful for us to improve the quality of this manuscript. We have carefully addressed all fruitful comments in the revised manuscript. We feel that the revised manuscript is of considerably improved quality and hope that it can meet the increasingly high-quality standard of this journal. 

Reviewer 3 Report

The authors addressed a number of concerns by improving their presentation and including new results regarding classification of strong and weak enhancers.  It is still not satisfying that the authors did not attempt to use cross-validation strategies nor use a different external data set to validate their model. I would like to ask the authors to consider these changes. 
